# DNA Damage Regulates Senescence-Associated Extracellular Vesicle Release via the Ceramide Pathway to Prevent Excessive Inflammatory Responses

**DOI:** 10.3390/ijms21103720

**Published:** 2020-05-25

**Authors:** Kazuhiro Hitomi, Ryo Okada, Tze Mun Loo, Kenichi Miyata, Asako J. Nakamura, Akiko Takahashi

**Affiliations:** 1Project for Cellular Senescence, The Cancer Institute, Japanese Foundation for Cancer Research, Koto-ku, Tokyo 135-8550, Japan; kazuhiro.hitomi@jfcr.or.jp (K.H.); ryo.okada@jfcr.or.jp (R.O.); tzemunloo@jfcr.or.jp (T.M.L.); kenichi.miyata@jfcr.or.jp (K.M.); 2Graduate School of Science and Engineering, Ibaraki University, Mito, Ibaraki 310-8512, Japan; asako.nakamura.wasabi@vc.ibaraki.ac.jp; 3Precursory Research for Embryonic Science and Technology (PRESTO), Japan Science and Technology Agency (JST), Kawaguchi, Saitama 332-0012, Japan; 4Advanced Research & Development Programs for Medical Innovation (PRIME), Japan Agency for Medical Research and Development (AMED), Chiyoda-ku, Tokyo 104-0004, Japan

**Keywords:** DNA damage, extracellular vesicle (EV), exosome, ceramide pathway, cellular senescence, senescence-associated secretory phenotype (SASP), senescence-associated extracellular vesicle (SA-EV), autophagy, bacterial infection, Bacillus Calmette–Guérin (BCG)

## Abstract

DNA damage, caused by various oncogenic stresses, can induce cell death or cellular senescence as an important tumor suppressor mechanism. Senescent cells display the features of a senescence-associated secretory phenotype (SASP), secreting inflammatory proteins into surrounding tissues, and contributing to various age-related pathologies. In addition to this inflammatory protein secretion, the release of extracellular vesicles (EVs) is also upregulated in senescent cells. However, the molecular mechanism underlying this phenomenon remains unclear. Here, we show that DNA damage activates the ceramide synthetic pathway, via the downregulation of sphingomyelin synthase 2 (SMS2) and the upregulation of neutral sphingomyelinase 2 (nSMase2), leading to an increase in senescence-associated EV (SA-EV) biogenesis. The EV biogenesis pathway, together with the autophagy-mediated degradation pathway, functions to block apoptosis by removing cytoplasmic DNA fragments derived from chromosomal DNA or bacterial infections. Our data suggest that this SA-EV pathway may play a prominent role in cellular homeostasis, particularly in senescent cells. In summary, DNA damage provokes SA-EV release by activating the ceramide pathway to protect cells from excessive inflammatory responses.

## 1. Introduction

DNA damage can have numerous chemical and biological effects on cellular function and may induce apoptotic cell death or cellular senescence in healthy cells [1,2]. Cellular senescence is an important tumor suppression mechanism. It acts as a barrier against various oncogenic stresses, such as telomere shortening, oncogene activation, irradiation, or stimuli that damage DNA [3,4,5]. These stressors activate p53, followed by an increase in p21^WAF1/CIP1^ and/or p16^INK4a^ expression. These are cyclin-dependent kinase inhibitors that promote the activation of retinoblastoma protein (RB), resulting in irreversible cell cycle arrest in normal cells [6,7,8]. Based on in vivo imaging analysis, using p21^WAF1/CIP1^ or p16^INK4a^ transgenic mice, we found that senescent cells accumulated throughout their bodies during the aging process [9,10,11]. Recently, it was shown that senescent cells increase the expression of inflammatory genes and produce various secretory proteins, such as cytokines, chemokines, growth factors, and matrix metalloproteinases [12,13]. This phenotype is referred to as the senescence-associated secretory phenotype (SASP) and is known to cause chronic inflammation in surrounding tissues [5,14,15]. SASP is associated with multiple age-related pathologies, such as cancer [16]. SASP-factor gene expression is transcriptionally regulated by nuclear factor κB (NF-κB) or CCAAT/enhancer-binding protein β (C/EBP-β) in senescent cells [17]. In addition, DNA damage signaling is important for SASP-factor gene expression, both through the epigenetic modification of chromatin [18] and by activation of the innate immune response via the cyclic GMP-AMP synthase (cGAS)-stimulator of interferon genes (STING) pathway after chromosomal DNA fragments accumulate in the cytoplasm [19,20,21,22,23].

Notably, recent evidence has shown that senescent cells secrete not only inflammatory proteins, but also small extracellular vesicles (EVs) [24,25,26], which contain various cellular components, such as proteins, lipids, and nucleic acids, and thereby play a role in cell-to-cell communication [27,28,29]. Small EVs, released from senescent cells, contribute to cancer cell proliferation and the induction of cellular senescence in neighboring cells, as so-called SASP factors [16,25,30,31]. In addition, we previously reported that small EV secretion is an essential defense mechanism for avoiding the aberrant activation of DNA damage responses in normal cells and for maintaining cellular homeostasis by the excretion of toxic self-DNA or viral DNA into the extracellular space [24]. However, the detailed molecular mechanism of EV biogenesis, triggered by DNA damage signaling in senescent cells, has yet to be elucidated fully. In this study, we demonstrated that DNA damage promotes activation of the ceramide pathway and increases the release of small EVs. Furthermore, this pathway, in conjunction with autophagy, prevents excessive inflammatory responses against bacterial infection.

## 2. Results

### 2.1. DNA Damage Induces Small EV Secretion from Normal Human Fibroblasts and Epithelial Cells

To evaluate the effect of DNA damage on small EV secretion from normal human diploid fibroblasts (HDFs), TIG-3 cells were treated with a DNA-damaging agent, doxorubicin (DXR). Treatment with DXR increased the signs of the DNA damage response (DDR) in a dose-dependent manner, as judged by DNA damage foci (the phosphorylation of H2AX and the consensus target sequences of ATM/ATR) (Figure 1A). Consistent with several previous reports [24,26,32,33], nanoparticle tracking analysis (NTA) revealed that an increase in small EV secretion was concomitant with the level of DNA damage (Figure 1B). Since DNA damage is known to cause cellular senescence in normal cells, we investigated the molecular mechanism of small EV secretion, induced by DNA damage, using both young and senescent HDFs. We confirmed the induction of cellular senescence by RT-qPCR analysis of p16^INK4a^, a well-established marker of cellular senescence, in DNA damage- or oncogene (HRasV12)-induced senescent cells (Figure 1C) [6,7,8]. Recent studies have shown that there are many factors involved in regulating the biogenesis and release of small EVs, such as the ceramide pathway, ESCRT (endosomal sorting complexes required for transport), and Rab-family small GTPases [34,35,36,37]. Of these, we focused on lipid-related proteins because the sphingolipid pathway might modulate the budding of intraluminal membrane vesicles (ILVs) into multivesicular bodies (MVBs) and regulate small EV biogenesis [34,38]. Here, we revealed that the gene expression of sphingomyelin synthase 2 (SMS2) was downregulated significantly while, conversely, that of neutral sphingomyelinase 2 (nSMase2) was upregulated in both types of senescent cells (Figure 1C,D). Sphingomyelinase activation and sphingomyelin synthase inhibition result in ceramide production, which is related to the promotion of small EV biogenesis and increased EV release from several cell types [34,38,39,40,41]. Therefore, we speculated that activating the ceramide synthetic pathway after DNA damage might be important for small EV release from both healthy control and senescent cells. To confirm our findings in another cell line, human retinal pigment epithelial cells (RPE-1 cells) were also treated with DXR to induce cellular senescence. In accordance with DDR induction and p16^INK4a^ upregulation (Figure 1E, F), the gene expression of both SMS2 and nSMase2 in the epithelial cells was changed in a similar manner to that seen in senescent HDFs (Figure 1F,G). Next, small EVs were subject to NTA, demonstrating that the release of small EVs also increased in senescent epithelial cells, compared with their release in healthy control cells (Figure 1H). In addition, transmission electron microscopy analysis using immuno-gold labelling for CD63, a well-known exosome marker [42], showed that these cells were secreting exosomes among the small EVs (Figure 1I). These data indicated that DNA damage activates the ceramide synthetic pathway in both HDFs and epithelial cells.

### 2.2. Activation of the Ceramide Synthetic Pathway Promotes Small EV Release from Cells

The expression levels of both SMS2 and nSMase2 changed in senescent cells; therefore we investigated these proteins’ roles in small EV release from HDFs. First, we used small interfering RNA (siRNA) to knock-down SMS2 [43], causing a significant induction of small EV secretion from HDFs, as determined by NTA (Figure 2A–C). Conversely, SMS2 overexpression reduced the level of small EV secretion after DXR treatment (Figure 2D,E). Second, nSMase2 depletion substantially reduced small EV secretion (Figure 2F–H) [38]. Importantly, inhibiting small EV secretion provoked the aberrant activation of DNA damage signaling in normal HDFs, as previously reported (Figure 2I) [24]. Furthermore, nSMase2 overexpression resulted in remarkably enhanced small EV release (Figure 2J,K). Taken together, these results revealed that activating the ceramide synthetic pathway promotes the release of small EV from cells.

### 2.3. Small EV Release Via the Ceramide Pathway Prevents DNA Damage Accumulation in Mice

In order to examine the effect of the ceramide synthetic pathway on both small EV release and tissue homeostasis in vivo, we used a chemical inhibitor of nSMase, spiroepoxide, which blocks small EV production in human cells [24,41]. We also observed the same effects in mouse embryonic fibroblasts (MEFs) by spiroepoxide treatment (Figure 3A). It is notable that inhibiting the ceramide pathway clearly induced cell cycle arrest and DNA damage accumulation in MEFs (Figure 3B,C). Next, we treated mice with spiroepoxide for 14 days. As expected, the inhibitor treatment reduced small EV release from the small intestine and accumulated DNA damage in mice tissues (Figure 3D,E). Collectively, our data strongly suggested that the ceramide pathway plays a crucial role in vivo in releasing small EVs and maintaining tissue homeostasis to avoid DNA damage accumulation.

### 2.4. The Autophagy Pathway Prevents Accumulation of Chromosomal DNA Fragments in the Cytoplasm to Block SASP-Factor Gene Expression in Accompany with Small EV Release

The accumulation of chromosomal DNA fragments in the cytoplasm promotes aberrant activation of the DNA-sensing pathway and causes global SASP-factor gene expression, both of which are associated with chronic inflammation and various age-related pathologies [15,16]. Cytoplasmic DNA fragments are so dangerous that normal cells prevent their accumulation via several degradation mechanisms, such as cytoplasmic DNases and the autophagy pathway [44]. We previously reported that the expression of the cytoplasmic DNases, DNase2 and TREX1, is regulated by E2F transcription factors and thereby downregulated by DNA damage [22]. In addition, the small EV secretion pathway also removes harmful cytoplasmic DNA fragments from cells [24,45]. However, the relationship between small EV release and autophagy in the regulation of cytoplasmic DNA fragments has remained unclear. Previous reports indicated that activating ceramide pathway blocks the autophagy-mediated degradation pathway [46]. Therefore, we speculated that DNA damage might block the autophagy pathway through ceramide pathway activation and, in turn, promote small EV release to prevent the accumulation of chromosomal DNA fragments in the cytoplasm of normal cells. We treated TIG-3 cells with autophagy pathway inhibitors that target lysosomes, chloroquine (CQ) and bafilomycin A1 (BafA1) [47]. Autophagy pathway inhibition caused an accumulation of chromosomal DNA fragments in the cytoplasm and cell cycle arrest in normal HDFs (Figure 4A–C). Significantly, the gene expression of a number of SASP factors was upregulated by these inhibitors (Figure 4D). Intriguingly, these treated cells also showed increases in the release of small EVs and double strand DNA (dsDNA) fragments from cells (Figure 4E,F). Similar to the situation in HDFs, DNA damage enhanced SASP-factor gene expression and small EV release in *ATG5* knockout cells compared with their expression and release, respectively, in wild-type cells (WT) (Appendix A) [48]. Research has previously shown that autophagy dysregulations can be detected in aged cells, and autophagy activation tended to improve senescent phenotypes [49,50]. Therefore, we treated senescent HDFs with rapamycin, an mTOR signaling inhibitor and autophagy inducer [51,52,53]. Strikingly, rapamycin treatment blocked IFN-β and CXCL10 expression in senescent cells (Appendix A). Taken together, these data strongly indicated that downregulation of the autophagy pathway promotes small EV release for the secretion of harmful chromosomal DNA fragments from cells (Figure 4G).

### 2.5. Small EV Release and Autophagy Cooperatively Prevent Cell Death Caused by Bacterial Infection

Previously, we reported that small EV secretion prevented adenovirus infection by excluding viral DNA from cells [24]. Additionally, we assessed the interplay between the release of small EVs and autophagy in preventing inflammation caused by bacterial infection. After differentiation into mature macrophage-like cells by Phorbol 12-myristate 13-acae-tate (PMA) stimulation for 7 days, human monocytic leukemia cells (THP-1 cells) were then infected with Bacillus Calmette–Guérin (BCG) vaccine, which is an attenuated form of *Mycobacterium bovis*, with or without inhibiting small EV biogenesis or secretion using siRNA oligos against Alix or Rab27a, as previously described (Figure 5A,B) [24]. Bacterial infection activates inflammatory responses in host cells. Indeed, inhibiting the small EV pathway dramatically increased the expressions of inflammatory genes, such as *IFN-β* and *CXCL10* (Figure 5C), resulting in apoptotic cell death (Figure 5D). Notably, the cleaved forms of both Caspase 3 and Caspase 1, markers of inflammasome activation, were clearly detected by the inhibition of small EV secretion in THP-1 cells following infection with BCG (Appendix A). Since bacterial DNA activates inflammasomes via the cGAS-STING pathway [54,55,56,57], STING depletion using previously validated siRNA oligos blocked both inflammatory gene expression and apoptotic cell death, suggesting that small EV release prevents excessive inflammatory responses caused by STING activation. The autophagy pathway also acts as a barrier against bacterial infection [58]. The levels of bacterial genomic DNA in host cells and small EV release increased considerably when autophagy was inhibited by adding 3-MA, an autophagy inhibitor (Appendix A) [59]. Importantly, apoptotic cell death increased significantly by inhibiting both autophagy and EV biogenesis (Appendix A). Collectively, these results demonstrate that small EV release and autophagy function cooperatively to prevent cellular inflammatory responses against bacterial infection.

## 3. Discussion

Senescent cells are metabolically active in a state of stable cell cycle arrest; they accumulate in the living body during the aging process [9,10,11]. These cells have been reported to play both beneficial and deleterious roles in our health through secreting SASP factors [12,13,14,15,16,17], and they also release many types of EVs, such as exosomes, microvesicles, nucleosomes, and apoptotic bodies, which are characterized by their size and secretory machinery [60,61,62,63]. Previously, we and other groups have reported that small EVs are actively released from senescent cells, functioning as harmful SASP factors by regulating the growth and viability of cancer cells [24,25,31]. According to recent studies, the biological functions of small EVs released from senescent cells change drastically because of changes in the composition of their proteins, lipids, and nucleic acids during cellular senescence [25,26,60,62,63,64]. In this study, we demonstrated that DNA damage is a key trigger, not only of senescence induction but also of small EV biogenesis via ceramide synthesis in senescent cells. The most common downstream mediator of DNA damage signaling, p53, reportedly induces the expression of several genes involved in endosome regulation and promote small EVs production [65]. In agreement with a previous report, we observed that DNA damage increased nSMase2 expression in HDFs and epithelial cells, suggesting that p53 activation might be involved in this phenomenon [66]. Additionally, we discovered that the expression of SMS2, an antagonistic enzyme of nSMase2 ceramide synthesis, reduced dramatically during cellular senescence (Figure 1C,D,F,G). The activation of sphingolipid-metabolizing enzymes is important for clearing amyloid beta (Aβ) protein from the brain is associated with preventing Alzheimer’s disease [40]. Therefore, it would appear to be important to reveal the mechanism involved in regulating SMS2 expression by DNA damage, and further investigations are required.

Cytoplasmic DNA fragments, derived from endogenous chromosomal or mitochondrial DNA resulting from DNA damage or from exogenous viral or bacterial infection, are detected by specific DNA-sensing machinery, such as the cGAS-STING pathway. Recent studies have shown that the cytoplasmic DNA-sensing pathway is critical for SASP-factor gene expression in senescent cells [19,20,21,22,23]. The innate immune response is important in fighting viral or bacterial infections, although excessive activation of this pathway can be dangerous for cells because the sustained activation of inflammasomes can result in additional DNA damage and/or cell death. Therefore, both DNA degradation enzymes and autophagy work to clear harmful cytoplasmic DNA fragments. However, in senescent cells, these cellular fail-safe mechanisms may not function normally because of persistent DNA damage signaling. Thus, the EV-mediated secretion pathway may also play a role in clearing cytoplasmic DNA fragments (see the model in Figure 6).

Altogether, we have elucidated that DNA damage can provoke senescence-associated EV (SA-EV) secretion by activating the ceramide pathway, thereby helping to eliminate hazardous DNA fragments from cells. It is notable that there might be alternative mechanisms for SA-EV secretion, and these could be potential therapeutic targets for the prevention of SASP, along with senolytic drugs to eliminate senescent cells [67]. Therefore, we will conduct further studies to address the molecular mechanisms underlying SA-EV biogenesis.

## 4. Materials and Methods

### 4.1. Cell Culture

TIG-3 and RPE-1 cells were obtained from Japanese Cancer Research Resources Bank (JCRB, Osaka, Japan) and mouse primary fibroblasts (MEFs) were established from day 13.5 mouse embryos. Wild type (WT) and atg5-/- immortalized MEFs [48] were obtained from RIKEN Cell Bank (RCB2710 and RCB2711, respectively) (RIKEN BRC, Ibaraki, Japan). These cells were cultured in Dulbecco’s Modified Eagle’s (DME) medium (Nacalai Tesque, Kyoto, Japan) supplemented with 10% fetal bovine serum (FBS). THP-1 cells were obtained from Cell Resource Center for Biomedical Research, Tohoku University. They were cultured in RPMI 1620 medium (Nacalai Tesque) supplemented with 10% FBS. In this research, young control cells were less than 40 population doublings and replicative senescent cells were more than 70 population doublings. TIG-3 cells were infected with recombinant retroviruses encoding Ras^V12^ (in pBabe–puro) or SMS2 or nSMase2 (in pMarX–puro) cDNA and treated with puromycin for 7 days [22]. For senescence induction, cells were irradiated with 10 Gy or 20 Gy X-rays using CP-160 (Faxitron X-ray Inc., Arizona, USA). An nSMase inhibitor GW4869 (Sigma-Aldrich, St. Louis, MO, USA), spiroepoxide (Santa Cruz, Dallas, TX, USA), doxorubicin (Wako, Osaka, Japan), chloroquine (Sigma-Aldrich), bafilomycin A1 (Sigma-Aldrich), rapamycin (Calbiochem, Merck, Darmstadt, Germany), PMA (Promega, Madison, WI, USA) or 3-MA (Calbiochem) were used in some experiments. Bacillus Calmette-Guérin (BCG) was purchased from Japan BCG Laboratory (Japan BCG Laboratory, Tokyo, Japan). All culture cells were confirmed the absence of mycoplasma contamination.

### 4.2. Cell Proliferation Assay

Cells were plated on 35 mm dishes with 2 mm grids (Thermo Fisher Scientific, Waltham, MA, USA). The number of cells in each grid was counted every day, and the relative number of cells was calculated based on an adjusted cell number at day 1 set at 1.0 as described previously [7].

### 4.3. Apoptosis Assay

Apoptotic cells were judged by FITC-Annexin V staining using an apoptotic/healthy cells detection kit (PromoKine, Heidelberg, Germany) as described previously [24]. After an incubation at room temperature for 15 min, fluorescence signals were measured with a Wallac ARVO 1420 Multilabel counter (PerkinElmer Co., Ltd., Waltham, MA, USA).

### 4.4. Cytoplasmic Nuclear DNA Analysis

Cytoplasmic DNA was prepared by modifying a method as previously reported [22,68]. In brief, cells were centrifuged for 1 min in a microcentrifuge, then suspended in 0.3 M sucrose buffer and homogenized with pipetting. The homogenate was overlaid on the same amount of 1.5 M sucrose buffer and centrifuged at 13,200 rpm for 10 min. Cytoplasmic DNA was purified by 0.4 mg/mL Proteinase K (Wako) treatment, phenol/chloroform extraction and ethanol precipitation with a carrier (Dr. GenTLE^®^, Takara Bio, Shiga, Japan). Cytoplasmic nuclear DNA was estimated by quantitative Real-Time PCR, using three different sets of primers as follows: human *GRM7*, 5′-TCAAGTGCCACATCCTATGC-3′ (forward), and 5′-ATTTTTCTAGCCAGGCACCA-3′ (reverse); human *FGFR2*, 5′-ACCTGGAAATGGCTGAAATG-3′ (forward), and 5′-AAGTCCTCGCAGAGGTTTCA-3′ (reverse); human *GPC6*, 5′-CGCCAGTGTGTGTAGCACTT-3′ (forward), and 5′-TCGGCCTCTCTCAGTTCTGT-3′ (reverse) [22].

### 4.5. Small Extracellular Vesicle Isolation from Cells

Small EVs were obtained from cell supernatants, as previously described with some modifications [24,27,69]. In brief, cells were incubated in DME medium with ultracentrifuged 5% FBS for 48 hours. The supernatants were collected and centrifuged at 300 *g* for 5 min and then at 2000× *g* for 10 min. The supernatant was then centrifuged at 10,000× *g* for 30 min, followed by filtration through a 0.2-μm pore filter (17597K, Sartorius, Gottingen, Lower Saxony, Germany). The collected supernatant was then subjected to preparation procedures by either ultracentrifugation at 100,000× *g* for 70 min as previously described [24,25] or an affinity-based method using MagCapture (Fujifilm Wako Chemicals, Tokyo, Japan) [70] for small EVs isolation. Nanoparticle tracking analysis were performed using a NanoSight LM10 system (Malvern Panalytical, Westborough, MA, USA). The amount of dsDNA in EVs was determined by quantitative Real-Time PCR using these primers: human *LINE1*, 5′-CAAACACCGCATATTCTCACTCA-3′ (forward), and 5′-CTTCCTGTGTCCATGTGATCTCA-3′ (reverse) [23,24].

### 4.6. Fluorescence Microscopic Analysis

The cells were fixed with 4% paraformaldehyde/PBS (Wako) and permeated through the membrane with 0.5% Triton X-100/tris-buffered saline (TBS) for 1 minutes. The cells were blocked with 1% bovine serum albumin (BSA) and 10% goat serum/TBS for 1 hour at 4 °C. Then, it was made to react with the following antibodies: γH2AX (1: 1,000, 05-636, Millipore, Temecula, CA, USA), phospho-(Ser/Thr) ATM/ATR substrate (1: 500, 2851, Cell Signaling Technology, Danver MA, USA), 53BP1 (1: 250, sc-58749, Santa Cruz). The secondary antibody was reacted using Alexa488 and Alexa594 (Thermo Fisher Scientific), and the nucleus was stained using DAPI (Dojindo, Tokyo, Japan). After immunostaining, DNA damage-positive cells were quantified by observation using a fluorescence microscope (Carl Zeiss, Oberkochen, Germany).

### 4.7. Electron Microscopy

Small EVs isolated from RPE-1 cells were absorbed to formvar carbon coated nickel grids and immune-labelled with an anti-CD63 antibody (556019, BD Biosciences, NJ, USA), followed by 5 nM of a gold-labelled secondary antibody (British BioCell International Ltd., UK). The samples were fixed in 2% glutaraldehyde in 0.1 M phosphate buffer. The grids were placed in 2% glutaraldehyde in 0.1 M phosphate buffer and dried, then stained with 2% uranyl acetate for 15 min and a Lead stain solution (Sigma-Aldrich). The samples were observed with a transmission electron microscope (JEM-1400Plus, JEOL Ltd., Tokyo, Japan) at 80 kV. Digital images were obtained with a CCD camera (VELETA, Olympus Soft imaging solutions GmbH, Olympus, Tokyo, Japan) [24].

### 4.8. RNAi

The sequences of the siRNA oligos were as follows. SMS2 [43]: GGGCAUUGCCUUCAUAUAU. nSMase2 [38]: GGAGGUGUUUGACAAGCG. Alix [24,36,38]: GAACCUGGAUAAUGAUGAA. Rab27a [24,35]: GCUGCCAAUGGGACAAACA. ON-TARGETplus siRNAs to target Sting mRNA sequences and non-targeting control siRNA were used (Dharmacon, Waterbeach, Cambridge, UK) [22,71].

### 4.9. Plasmids

The epitope tagged cDNAs of SMS2 and nSMase2 was cloned into the pMarX-puro retrovirus vector. All cDNAs were sequenced on a Genetic Analyzer 3130 (Applied Biosystems, Waltham, MA, USA) using a BigDye Terminator v3.1 Cycle Sequencing Kit (Applied Biosystems).

### 4.10. Quantitative Real-Time PCR

Total RNA was prepared using a mirVana kit (Thermo Fisher Scientific), and then subjected to reverse transcription using a PrimeScript RT reagent kit (Takara Bio Inc., Shiga, Japan). The expression levels of each mRNA were examined by quantitative real-time RT-PCR on a StepOnePlus PCR system (Applied Biosystems) using SYBR Premix Ex Taq (Takara Bio Inc.). The PCR primer sequences used are as follows: human *GAPDH*, 5′-CAACTACATGGTTTACATGTTC-3′ (forward) and 5′-GCCAGTGGACTCCACGAC-3′ (reverse) [22]; human p16 (*cdkn2a*), 5′-CGAATAGTTACGGTCGGAGG-3′ (forward) and 5′-TGAGAGTGGCGGGGTCG-3′ (reverse) [22]; human p21 (*cdkn1a*), 5′-TCAGGGTCGAAAACGGCG-3′ (forward) and 5′-AAGATCAGCCGGCGTTTGGA-3′ (reverse) [18]; *SMS2*, 5′-AGGAGCTTAGCCCTCCACTC-3′ (forward) and 5′-AACAGAATCTGCGTCCCACT-3′ (reverse); *nSMase2*, 5′-CAACAAGTGTAACGACGATGCC-3′ (forward) and 5′-GGCATCGTCGTTACACTTGTTG-3′ (reverse) [72]; human *IL-6*, 5′-CCAGGAGCCCAGCTATGAAC-3′ (forward) and 5′-CCCAGGGAGAAGGCAACTG-3′ (reverse) [22]; *IL-8*, 5′-AAGGAAAACTGGGTGCAGAG-3′ (forward) and 5′-ATTGCATCTGGCAACCCTAC-3′ (reverse) [22]; human *IL-1*α, 5′-AACCAGTGCTGCTGAAGGA-3′(forward) and 5′-TTCTTAGTGCCGTGAGTTTCC-3′ (reverse) [22]; human *IL-1*β, 5′-CTGTCCTGCGTGTTGAAAGA-3′ (forward) and 5′-TTGGGTAATTTTTGGGATCTACA-3′ (reverse) [22]; human *IFN-β*, 5′-AAACTCATGAGCAGTCTGCA-3′ (forward) and 5′-AGGAGATCTTCAGTTTCGGAGG-3′ (reverse) [22]; human *CXCL10*, 5′-CCAGAATCGAAGGCCATCAA-3′ (forward) and 5′-CATTTCCTTGCTAACTGCTTTCAG-3′ (reverse) [22]; human *Alix*, 5′-CCCAAATTCCCATTTTCTGA-3′ (forward) and 5′-GAGCCTCCAAAAAGTGAACC-3′ (reverse) [24]; human *Rab27a*, 5′-ATCACAACAGTGGGCATTGA-3′ (forward) and 5′-CCCTGCTGTGTCCCATAACT-3′ (reverse) [24]; human *STING*, 5′-ATATCTGCGGCTGATCCTGC-3′ (forward) and 5′-GGTCTGCTGGGGCAGTTTAT-3′ (reverse) [19]; mouse *GAPDH*, 5′-CAACTACATGGTCTACATGTTC-3′ (forward) and 5′-CGCCAGTAGACTCCACGAC-3′ (reverse) [22]; mouse *p16*, 5′-GAACTCTTTCGGTCGTACCC-3′ (forward) and 5′-CGAATCTGCACCGTAGTTGA-3′ (reverse) [22]; mouse *IL-6*, 5′-CCGGAGAGGAGACTTCACAG-3′ (forward) and 5′-TCCACGATTTCCCAGAGAAC-3′ (reverse) [22]; mouse *GRO-α*, 5′-CCGCTCGCTTCTCTGTGC-3′ (forward) and 5′-CTCTGGATGTTCTTGAGGTGAATC-3′ (reverse) [18]; mouse *IFN-β*, 5′-CAGCTCCAAGAAAGGACGAAC-3′ (forward) and 5′-GGCAGTGTAACTCTTCTGCAT-3′ (reverse) [22]; mouse *CXCL10*, 5′-CCAAGTGCTGCCGTCATTTTC-3′ (forward) and 5′-GGCTCGCAGGGATGATTTCAA-3′ (reverse) [22]; *BCG*, 5′-GTCCACGCCGCCAACTACG-3′ (forward) and 5′-GTTAGGTGCTGGTGGTCCGAAG-3′ (reverse) [73]. The means ± s.d. of three independent experiments are shown.

### 4.11. Western Blotting

For Western blotting analysis, cells were lysed in lysis buffer (50 mM Hepes, pH 7.5, 150 mM NaCl, 1 mM EDTA, 2.5 mM EGTA, 10% glycerol, 0.1% Tween20, 10 mM β-glycerophosphate) with 1% Protease inhibitor cocktail (Nacalai Tesque). The protein concentration was determined using a Pierce™ BCA Protein Assay Kit (23225, Thermo Fisher Scientific), and the proteins were transferred to a PVDF membrane (EMD Millipore) after SDS-PAGE. Primary antibodies used in this study were anti-SMS2 (sc-366682, Santa Cruz), nSMase2 (sc-166637, Santa Cruz), H-Ras (sc-29, Santa Cruz), Alix (12422, Proteintech, Rosemont, IL, USA), Rab27a (17817, Proteintech), Caspase3 (9662, Cell Signaling Technology), Caspase1 (sc-1780, Santa Cruz), ATG5 (12994, Cell Signaling Technology), p62 (PM045, MBL International, Woburn, MA, USA), LC3 (2775, Cell Signal Transduction), α-tubulin (T9026, Sigma-Aldrich). After incubation with secondary antibodies (GE Healthcare, Chicago, IL, USA), membranes were treated with the SuperSignal West Femto Maximum Sensitivity Substrate (Thermo Fisher Scientific), and detected by FUSION SOLO S (Vilber Lourmat, Collegien, France).

### 4.12. Animal Experiments

CD1 (ICR) mice were purchased from Charles River Inc. (Wilmington, MA, USA). An N-SMase inhibitor, spiroepoxide (Santa Cruz), was intraperitoneally injected into 50-day-old male mice at 3.5 mg/kg every two day. Fourteen days later, mice treated with spiroepoxide were euthanized and the small intestine sections were subjected to small EV collection or immunofluorescence analysis as described previously [24]. All animal care was performed according to the protocols approved by the Committee for the Use and Care of Experimental Animals of the Japanese Foundation for Cancer Research (No.17-02-1, 31 5 2017).

### 4.13. Statistical Analysis

Statistical significance was determined using unpaired two-tailed Student’s *t*-test by Excell (Microsoft, Redmond, WA, USA). *p*-values less than 0.05 were considered significant.

## 5. Conclusions

We demonstrated how DNA damage activates the ceramide pathway and leads to increase senescence-associated EV (SA-EV) secretion. The SA-EV pathway is crucial for cellular homeostasis to protect cells from excessive inflammatory responses.

## Figures and Tables

**Figure 1 ijms-21-03720-f001:**
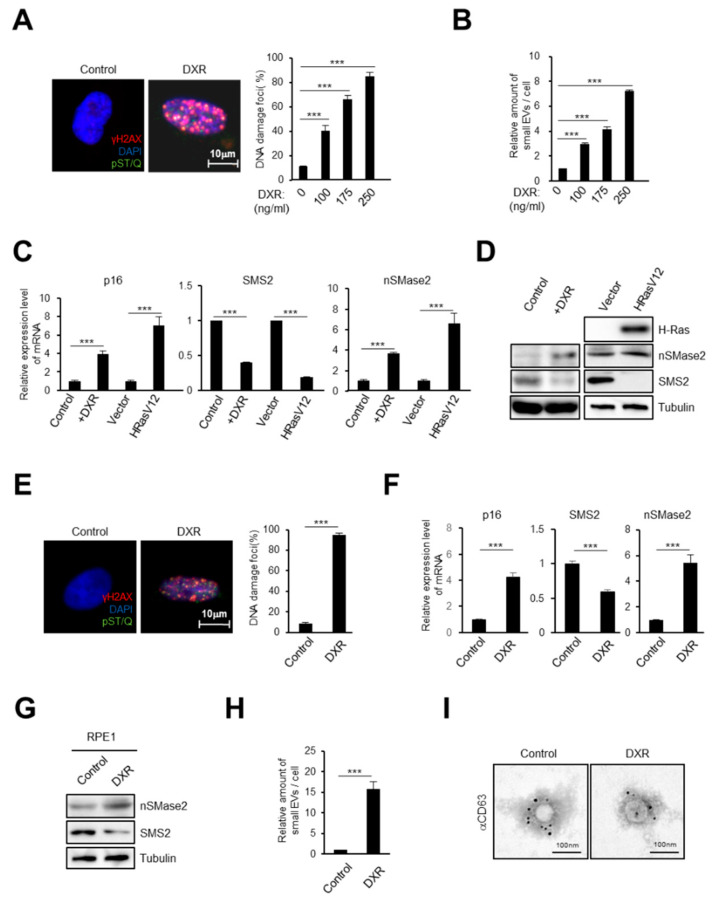
Gene expression associated with the ceramide synthetic pathway is changed by damage to DNA. (**A**,**B**) TIG-3 cells were treated with doxorubicin (DXR) for 48h and subjected to immunofluorescence staining for markers of DNA damage (γ-H2AX [red], phosphor-Ser/Thr ATM/ATR (pST/Q) substrate [green] and 40,6-diamidino-2-phenylindole: DAPI [blue]) (**A**) or to NanoSight analysis of isolated sEV particles (**B**). (**C**,**D**) Pre-senescent TIG-3 cells were rendered senescent by DXR treatment or retroviral infection of oncogenic *ras* (HRasV12), then subjected to RT-qPCR analysis for p16^INK4a^, SMS2 and nSMase2 gene expression (**C**) and western blotting (**D**). (**E**)–(**I**) Pre-senescent RPE-1 cells were treated with DXR and subjected to immunofluorescence staining for markers of DNA damage (γ-H2AX [red], pST/Q substrate [green] and DAPI [blue]) (**E**), RT-qPCR analysis (**F**) and to western blotting (**G**). The percentage of nuclei that contain more than 3 DNA damaging foci were shown in the histograms (**E**). NanoSight analysis of isolated sEV particles (**H**) and immuno-gold labelling for CD63, a well-known exosome marker, followed by transmission electron microscopy (TEM) (**I**). Scale bars, 10 μm. For all graphs, error bars indicate mean ± standard deviation (s.d.) of triplicate measurements. *p* values was calculated by unpaired two-tailed Student’s *t*-test (*** *p* < 0.001).

**Figure 2 ijms-21-03720-f002:**
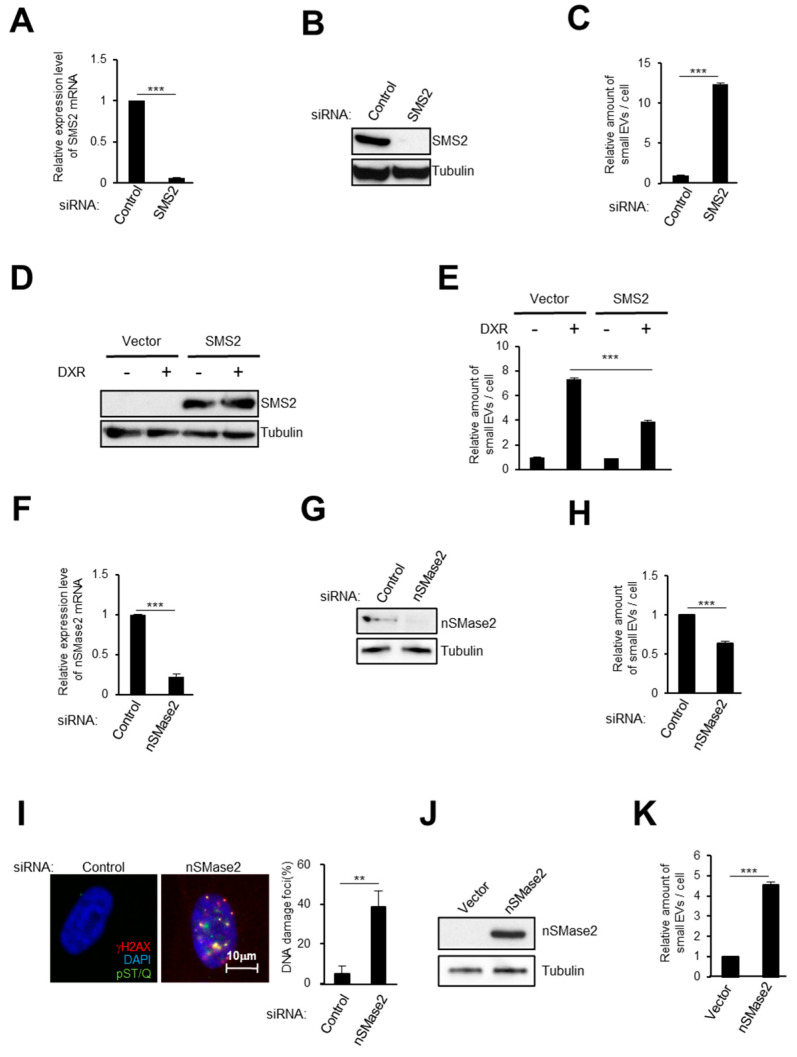
The ceramide pathway plays an important role in small EV secretion from HDFs. (**A**–**C**) After transfection with siRNA oligos against SMS2 twice, TIG-3 cells were then subjected to RT-qPCR analysis of SMS2 gene expression (**A**), western blotting (**B**), or to NanoSight analysis of isolated small EV particles (**C**). (**D**,**E**) After infection with retrovirus encoding FLAG-tagged SMS2 or empty vector and selection with puromycin, TIG-3 cells were treated with 150 nM DXR for 10 days and subjected to western blotting (**D**), or to NanoSight analysis of isolated small EV particles (**E**). (**F**–**H**) After transfection with siRNA oligos against nSMase2 twice, TIG-3 cells were subjected to RT-qPCR analysis of nSMase2 gene expression (**F**), western blotting (**G**), NanoSight analysis of isolated small EV particles (**H**), and to immunofluorescence staining for markers of DNA damage (γ-H2AX [red], pST/Q substrate [green] and DAPI [blue]) (**I**). The percentage of nuclei that contain more than 3 DNA damaging foci positive were shown in the histograms (**I**). (**J**,**K**) Pre-senescent TIG-3 cells were infected with retrovirus encoding FLAG-tagged nSMase2 or empty vector. After selection with puromycin, cells were subjected to western blotting (**J**), or to NanoSight analysis of isolated small EV particles (**K**). For all graphs, error bars indicate mean + standard deviation (s.d.) of triplicate measurements. *P* values was calculated by unpaired two-tailed Student’s *t*-test (** *p* < 0.01, *** *p* < 0.001).

**Figure 3 ijms-21-03720-f003:**
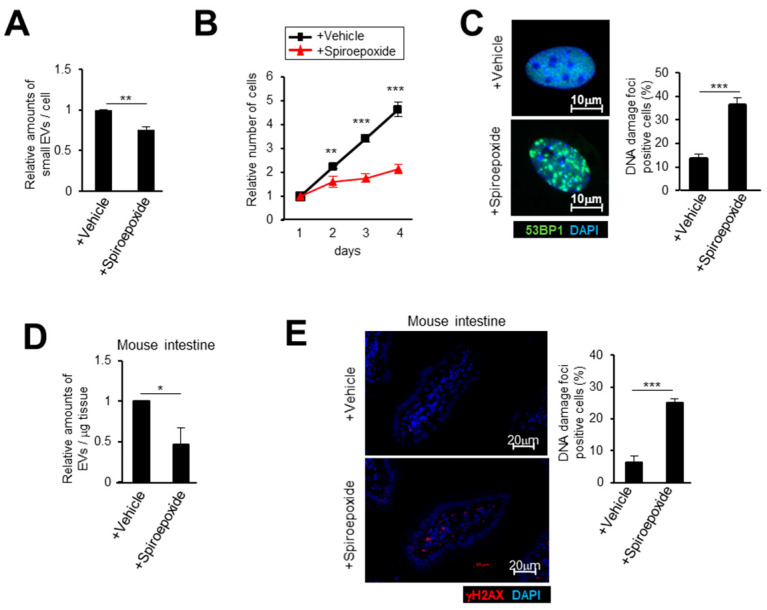
Inhibiting the ceramide pathway causes DNA damage accumulation in mice. (**A**–**C**) Mouse embryonic fibroblasts (MEFs) were treated with spiroepoxide and then subjected to NanoSight analysis of isolated small EV particles (**A**), cell proliferation analysis (**B**) or to immunofluorescence staining for markers of DNA damage (53BP1 [red] and DAPI [blue]) (**C**). Scale bars, 10 μm. The histograms indicate the percentage of nuclei that contain more than 3 foci positive for 53BP1 staining. At least 100 cells were scored per group (**C**). (**D**,**E**) ICR (CD1) mice were intraperitoneally injected with spiroepoxide every two days. After 14 days, the mice were euthanized and small intestines were subjected to NanoSight analysis (NTA) of isolated small EV particles (**D**) or to immunofluorescence analysis of intestine section (**E**). Section of intestines were subjected to immunofluorescence staining for markers of DNA damage (53BP1 [red] and DAPI [blue]) (**E**). The representative data from three independent experiments are shown. For all graphs, error bars indicate mean +standard deviation (s.d.) of triplicate measurements. *p* values was calculated by unpaired two-tailed Student’s *t*-test (* *p* < 0.05, ** *p* < 0.01, *** *p* < 0.001).

**Figure 4 ijms-21-03720-f004:**
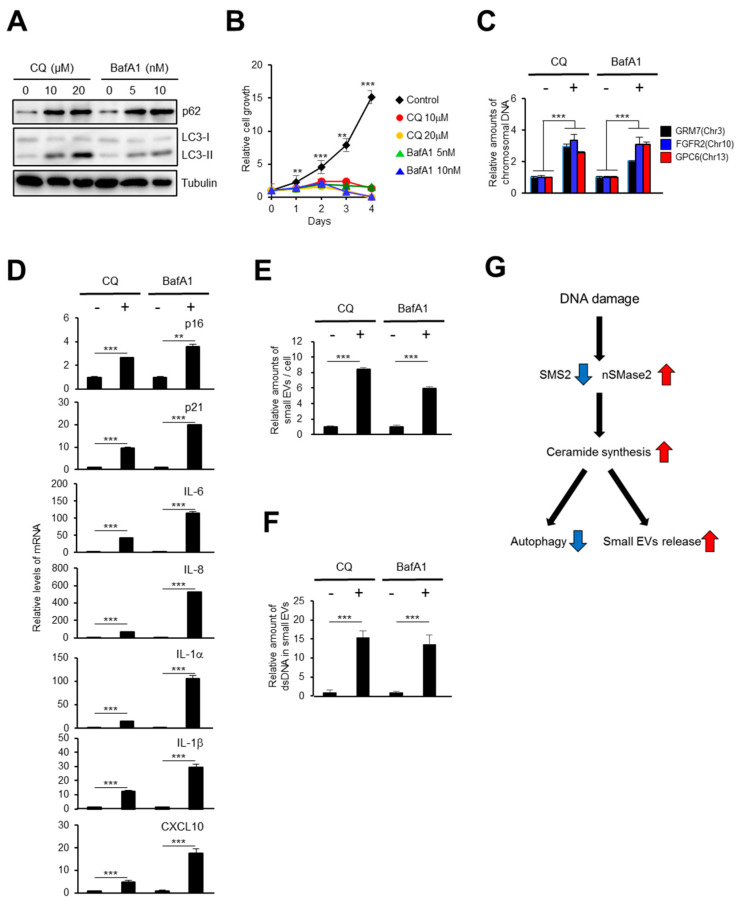
Inhibiting the autophagy pathway leads to the gene expression of SASP factors. (**A**,**B**) TIG-3 cells were treated with chloroquine (CQ) or bafilomycin A1 (BafA1) for 48 h and subjected to western blotting (**A**) and to cell proliferation analysis (**B**). (**C**–**F**) TIG-3 cells were treated with 20 μM Chloroquine or 10 nM bafilomycin A1 for 48 h and subjected to isolation of cytoplasmic fraction followed by qPCR analysis of chromosomal DNA in cytoplasm (**C**), RT-qPCR analysis of SASP factor gene expression (**D**), NanoSight analysis of isolated small EV particles (**E**) or to quantitative analysis of dsDNA in small EVs (**F**). (**G**) A model of the molecular mechanism. DNA damage activates the ceramide synthetic pathway, both through SMS2 downregulation and nSMase2 upregulation, blocking autophagy and promoting the release of small EVs. For all graphs, error bars indicate mean ± standard deviation (s.d.) of triplicate measurements. *p* values was calculated by unpaired two-tailed Student’s *t*-test (** *p* < 0.01, *** *p* < 0.001).

**Figure 5 ijms-21-03720-f005:**
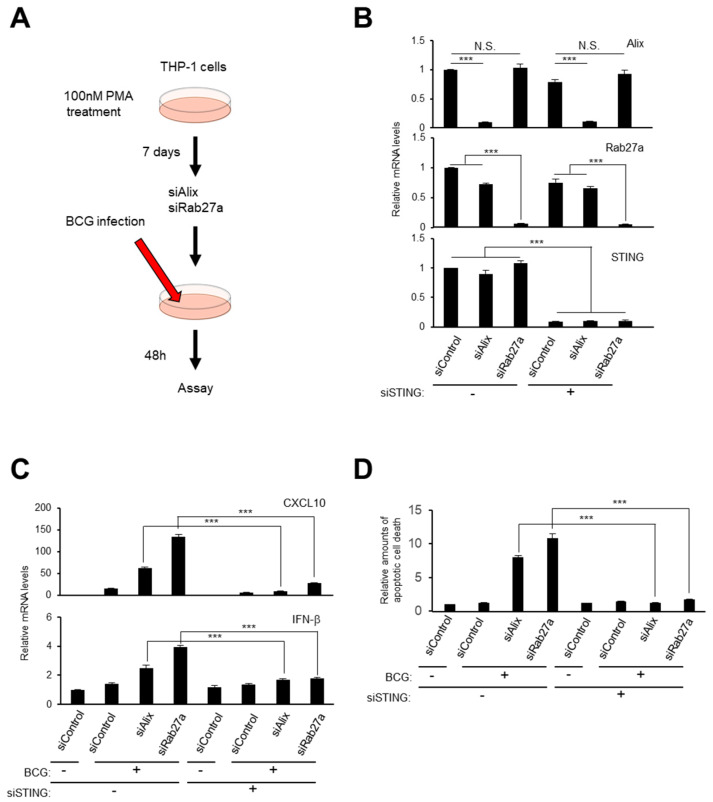
The release of small EVs blocks the inflammatory response caused by bacterial infection. (**A**) Timeline of the experimental procedure. (**B**–**D**) THP-1 cells were treated with 100 nM PMA for 7 days and transfected with indicated siRNA oligos. After incubation with Bacillus Calmette-Guérin (BCG), RT-qPCR analysis was performed for confirmation of knockdown efficiency (**B**) or innate immune responsible gene expression (**C**). The amounts of apoptotic cell death was quantitatively measured (**D**). For all graphs, error bars indicate mean +standard deviation (s.d.) of triplicate measurements. *p* values was calculated by unpaired two-tailed Student’s *t*-test (*** *p* < 0.001, N.S. (not significant)).

**Figure 6 ijms-21-03720-f006:**
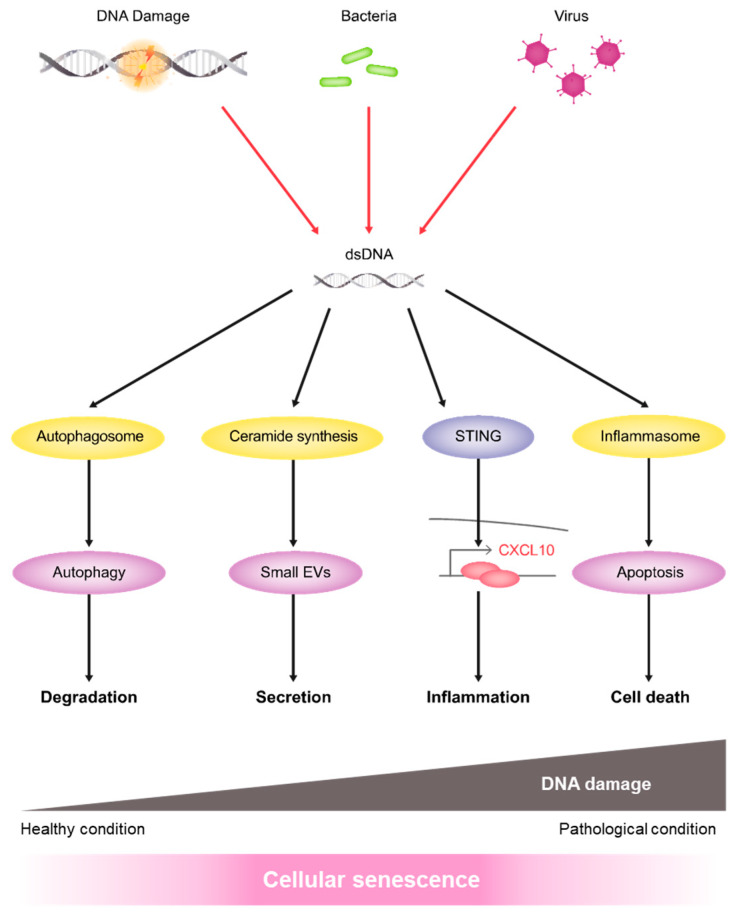
A model of our research. In healthy conditions, endogenous dsDNA fragments, derived from chromosomal or mitochondrial DNA, or exogenous dsDNA fragments, derived from bacterial or viral infection, are degraded by autophagy or released via small EVs. Therefore, these pathways function as defence mechanisms to prevent aberrant activation of innate immune responses. In senescent cells, irreparable DNA damage inhibits dsDNA degradation by autophagy and upregulates small EV secretion by activating the ceramide pathway. If neither pathway can function sufficiently, inflammasome activation occurs, leading to apoptosis. In summary, autophagy and the small EV pathway cooperatively play key roles to support cellular homeostasis.

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
