# Peer review of "DNA Damage Regulates Senescence-Associated Extracellular Vesicle Release via the Ceramide Pathway to Prevent Excessive Inflammatory Responses"

_ijms, 2020, doi:10.3390/ijms21103720_

Round 1

Reviewer 1 Report

Hitomi et al elucidated that DNA damage can provoke senescence-associated EV (SA282 EV) secretion by activating the ceramide pathway, thereby helping to eliminate hazardous DNA fragments from cells.

The work is scientifically sound and the authors have clearly presented their findings. However, I have a few concerns.

  1. I suggest characterizing the control and treatment derived exosomes according to MISEV guidelines. (https://www.tandfonline.com/doi/full/10.1080/20013078.2018.1535750)

  1. The authors stated that “Previous reports indicated that activating ceramide pathway blocks the autophagy-mediated degradation pathway. Therefore, we speculated that DNA 187 damage might block the autophagy pathway through ceramide pathway activation and, in turn, promote small EV release to prevent the accumulation of chromosomal DNA fragments in the cytoplasm of normal cells”.

 Some intraluminal vesicles undergo degradation by the lysosomal pathway and other ILVs get released as exosomes. I am just wondering what is the fate of these EVs which package chromosomal DNA fragments?  Do they transfer the packed chromosomal DNA fragment to other normal cells or they will be routed to some other pathway for the removal?

Reviewer 2 Report

Revision Manuscript ID ijms-803727

The authors of the manuscript entitled: DNA damage regulates senescence-associated extracellular vesicle release via the ceramide pathway to prevent excessive inflammatory responses” explore a new and interesting topic in medical research.

This field of study is very promising from a translational point of view because senescence is associated with tissue aging and age-related pathologies. Senescence, depending on the cellular context, could have an anti-cancer role for its anti-proliferative effects, or deleterious consequences, if associated with an inflammatory response (SASP) in tissues surrounding tumors.

In the present manuscript, authors explore the second phenotype: senescence-associated to inflammation in the context of DNA damage in different cellular models, normal human diploid fibroblasts, and a macrophage model (differentiated THP1) after bacterial infection. To confirm the in vitro results they used an animal model (CD1 ICR mice) too.

The authors focalize their study on the biochemical pathway of ceramide, associated with the new role of extracellular vesicles (EV) in the context of senescence and autophagy. Since the complex and still unknown contribution of these two phenomena in cellular homeostasis, this is a very interesting study with high scientific soundness and interest to the readers.

Methods and cell culture models are well described, results are clearly presented and excellently illustrated by useful schemes. The discussion is very interesting and convincing. The conclusion is supported by results, references are updated and appropriate. Finally, my overall recommendation is to accept the manuscript in the present form.

Author Response

We thank this reviewer for his/her fair assessment.